# Dating Whole Genome Duplication in *Ceratopteris thalictroides* and Potential Adaptive Values of Retained Gene Duplicates

**DOI:** 10.3390/ijms20081926

**Published:** 2019-04-19

**Authors:** Rui Zhang, Fa-Guo Wang, Jiao Zhang, Hui Shang, Li Liu, Hao Wang, Guo-Hua Zhao, Hui Shen, Yue-Hong Yan

**Affiliations:** 1Shanghai Chenshan Plant Science Research Center, Shanghai Chenshan Botanical Garden, Chinese Academy of Sciences, Shanghai 201602, China; zhangrui@csnbgsh.cn (R.Z.); zhangjiao@sibs.ac.cn (J.Z.); sh007-008@163.com (H.S.); 15609616871@163.com (L.L.); 1000441464@smail.shnu.edu.cn (H.W.); zhaoguohua1990@163.com (G.-H.Z.); shenhui@csnbgsh.cn (H.S.); 2Eastern China Conservation Center for Wild Endangered Plant Resources, Shanghai 201602, China; 3Key Laboratory of Plant Resources Conservation and Sustainable Utilization, South China Botanical Garden, Chinese Academy of Sciences, Guangzhou 510650, China; wangfg@scib.ac.cn

**Keywords:** whole genome duplication, *Ceratopteris thalictroides*, synonymous substitutions, transcriptome, evolution

## Abstract

Whole-genome duplications (WGDs) are widespread in plants and frequently coincide with global climatic change events, such as the Cretaceous–Tertiary (KT) extinction event approximately 65 million years ago (mya). Ferns have larger genomes and higher chromosome numbers than seed plants, which likely resulted from multiple rounds of polyploidy. Here, we use diploid and triploid material from a model fern species, *Ceratopteris thalictroides*, for the detection of WGDs. High-quality RNA-seq data was used to infer the number of synonymous substitutions per synonymous site (*Ks*) between paralogs; *Ks* age distribution and absolute dating approach were used to determine the age of WGD events. Evidence of an ancient WGD event with a *Ks* peak value of approximately 1.2 was obtained for both samples; however, the *Ks* frequency distributions varied significantly. Importantly, we dated the WGD event at 51–53 mya, which coincides with the Paleocene-Eocene Thermal Maximum (PETM), when the Earth became warmer and wetter than any other period during the Cenozoic. Duplicate genes were preferentially retained for specific functions, such as environment response, further support that the duplicates may have promoted quick adaption to environmental changes and potentially resulted in evolutionary success, especially for pantropical species, such as *C. thalictroides*, which exhibits higher temperature tolerance.

## 1. Introduction

Whole-genome duplication (WGD), or polyploidy, has long been considered as an important evolutionary force and often drives plant speciation [1,2,3,4,5]. With more than 300,000 living species, angiosperms are currently the largest group of land plants, and most flowering plants have experienced multiple rounds of WGD. Even the small genome of *Arabidopsis thaliana* has undergone two recent WGDs and one whole genome triplication during the course of its evolution [3,6,7]. Two WGD events (ρ and δ) are indicated to have occurred early in the monocot lineage after its divergence from the eudicot clade [2,3]. Furthermore, substantial evidence shows that ancient WGDs occurred in the common ancestor of extant seed plants, ferns, and mosses [3,5,8,9].

Massive WGDs in flowering plants have occurred at specific times during extreme environmental changes or extinction periods, including the Cretaceous–Tertiary (KT) boundary, and Paleocene-Eocene Thermal Maximum (PETM) [5,10,11]. WGDs are thought to be related to dramatic global climate changes and unstable environments [5,10,11,12]. Polyploidy could accelerate adaptation to dramatically changing environments via genetic innovations or heterotic effects, as well as mutational robustness, subfunctionalization, and changed modes of reproduction [10,11,13]. Interestingly, gene retention after WGDs is not random; a general pattern has been observed indicating that regulators and signal transducers are preferentially retained in vertebrates and flowering plants [3,14,15]. Another type of biased retention following WGDs are involved in response to biotic and abiotic stress in flowering plants and are important for local adaptation [5,15]. Most *MADS-box* family genes were retained after WGDs, especially *MIKC*-type genes [13,16]. Biased gene retention is considered to have an important impact on innovation and diversification that might contribute to long-term survival [5]. However, little is known about the occurrence of WGDs for nonflowering plants.

Ferns (Monilophyta) constitute the second largest group of vascular plants and an early diverged lineage of land plants; they possess a higher frequency of polyploidy than seed plants. Previous studies have reported that the frequency of polyploidy in ferns varies from 31%–95%, while that in flowering plants varies by 15%–70% [17,18]. With a high base chromosome number, ferns are prone to polyploidization, which might derive from ancient polyploidization [17,19]. The average chromosome number in homosporous ferns (*n* = 57.0) greatly exceeds that in angiosperms (*n* = 15.99) and heterosporous ferns (*n* = 13.62) [20]. Moreover, polyploid species are commonly more prevalent in nature than initially suspected [5]. Several WGDs in the ancestor of extant ferns were detected based on chromosome counts analysis [21,22]. Li et al. (2018) identified two WGDs using phylogenomic analyses and *Ks*-based age distribution: one *Azolla*-specific and one at the base of core leptosporangiates in *Azolla filiculoides* and *Salvinia cucullata* [23]. WGDs were also found in *Equisetum giganteum* and *Ceratopteris richardii* [9,24]. However, little is known about the whole genome sequences of ferns.

Recent improvement in technology, such as transcriptome sequencing, has allowed for a relatively convenient and efficient alternative to evaluate paleopolyploid events because a large number of sequences can be obtained at a low cost [24]. To determine polyploidy events, one such method requires determining the number of synonymous substitutions per synonymous site (*Ks*) between paralogs, which has been widely performed in lots of species [6,9,12]. Without changes in protein sequences, synonymous substitutions are considered to be putatively neutral and accumulate changes at a constant rate. Duplicated (paralogous) pairs in a genome can be sorted in order and time as substitutions accumulate over time by plotting the *Ks* age distribution; this data can be used to detect whether paleopolyploidizations have occurred in plants [6,9]. Most of these duplicates include small-scale duplication and recent neopolyploidizations that were lost frequently that result in an exponential decrease and create an initial peak in the young age classes, and a long and flat tail is expected for older duplicates over time due to fewer genes retained, resulting in an L-shaped pattern [9,25]. However, large-scale duplication events, such as WGDs, are expected to show a secondary peak superimposed upon this L-shaped distribution because of the burst of large-scale duplicates genes at about the same time [6,24]. A WGD is predicted if a clear peak is present in the age distribution of paralogous pairs. To obtain more insight into fern evolution, it is important to examine the genes of fern species to see if there are any signals that indicate multiple genome duplications (e.g., multiple *Ks* peaks), determine the time at which WGDs occurred, and identify whether gene retention is biased towards specific functional classes.

*Ceratopteris* is a pantropical genus of annual ferns. It is commonly used as a model for ferns because it is easy to culture and produces abundant spores. It also has an independent sporophyte, gametophyte generation, and short sexual cycle [26,27]. Here, we used two samples of *C. thalictroides* in previous published studies [28,29], performed chromosome counts on them, then detected WGD events using high-quality transcriptome data via *Ks* analysis. Additionally, we dated the WGD event using both *Ks* distribution peaks and the absolute dating approach. Furthermore, annotation of the biased gene retention following the WGD events was also analyzed. Our results will help improve the understanding of the genome history and evolutionary impact of WGD events in ferns.

## 2. Results

### 2.1. Chromosome Counting

As the genetic variation in a single sample fails to adequately represent the variation present within a species, we performed a cytological study on the root tips of two collected sporophytes to reflect their genetic background. The *C. thalictroides* sample reported by Shen et al. (2018) was identified as a diploid with a chromosome number of 2*n* = 78 (Figure 1A,B) [29]. The chromosome number of *C. thalictroides* published by Zhang et al. (2016) is 2*n* = 117 (Figure 1C,D), thus it is a possible triploid [28].

### 2.2. Benchmarking Universal Single Copy Orthologs (BUSCO) Analysis

A total of 74,728 and 69,929 contigs with an N50 of 1,610 bp and 787 bp, respectively, were obtained in the diploid and the triploid transcriptomes of *C. thalictroides*. A total of 83,202 and 60,823 unigenes from the diploid and the triploid transcriptomes of *C. thalictroides*, respectively, were then used to predict open reading frames with details provided in Table 1 [28,29].

We performed a BUSCO analysis to assess the level of completeness of the transcriptome assembly (Table 2) using a plant species database containing 1440 ortholog groups. We were able to identify 63.7% of complete single-copy genes and 5.9% partial sequences in the diploid. In addition, 46.5% of the BUSCO orthologs (12.2% of which were fragmented genes) were identified in the triploid. These results indicate that the data were of a high quality and can be used for subsequent analyses.

### 2.3. Identification of Pairs of Paralogous and WGD Events Estimated on the Basis of Ks Age Distributions

In this study, the distributions of *Ks* within paralogous pairs were determined to detect potential WGD events (Figure 2 and Table 3). The difference in *Ks* frequency distributions between the diploid and the triploid was significant (Wilcoxon matched pairs test; *p* = 0.035). A total of 8364 paralogous pairs were identified in the diploid *C. thalictroides*, whereas 3088 paralogous pairs were identified in the triploid sample. We used Gaussian mixture models to confirm the WGD signature. We observed seven and five peaks in the diploid and the triploid, respectively. A significant *Ks* peak near 1.2 was detected by the fitted Gaussian mixture models for both samples, and several other smaller peaks were also fitted. We only focused on *Ks* values less than 2 because higher *Ks* values are uncertain due to *Ks* saturation and stochastic effects, which might provide misleading data for the mixture models. Thus, one ancient WGD event with a median *Ks* peak at approximately 1.2 occurred in the diploid and triploid *C. thalictroides* samples, indicating that the large peak present in *C. thalictroides* was a true WGD rather than stochastic variation.

### 2.4. Functional Classifications of Retained Duplicates after WGD Events Revealed That Retained Genes Are Biased Rather Than Random

A gene ontology (GO) enrichment analysis was performed to explore the potential functions of retained duplicates following WGDs in the diploid with a higher quality for this study. The focus of the analysis was on the WGD event shared by the diploid and the triploid, with a *Ks* peak at approximately 1.2. The results showed that several GO terms were preferentially retained. A total of 111 GO terms were found to be significantly enriched (Appendix A). The following GO terms were found to be enriched: GO terms of type I genes that have major contributions to biological processes (BP), especially genes related to response to stimulus (response to bacterium, “GO:0009617”; cellular response to carbohydrate stimulus, “GO:0071322”; heat acclimation, “GO:0010286”; and cellular response to salt stress, “GO:0071472”) (Appendix A); GO terms of genes involved in signaling (signaling, “GO:0023052”; small GTPase-mediated signal transduction, “GO:0007264”; and sugar mediated signaling pathway, “GO:0010182”); and GO categories corresponding to various aspects of regulation (protein transport, “GO:0015031”; magnesium ion transport, “GO:0015693”; post-translational protein modification, “GO:0043687”; and protein amino acid autophosphorylation “GO:0046777”). Type II genes were related to cellular components (CC), such as cortical microtubules (GO:0055028) and Golgi apparatus (GO:0005794), and type III genes had specific functional roles that are associated with kinase activity, protein binding, transporter activity, and catalytic activity.

### 2.5. Dating the WGD Event in C. thalictroides Using Ks Distribution Peaks and Absolute Dating

We determined the age of the WGD events using both *Ks* distribution peaks and the absolute dating approach. Without changes of protein sequences, synonymous substitutions are considered to be putatively neutral and accumulate changes at a constant rate, which can be used to infer the age of WGDs. When a plant average *Ks*/year rate of 6.1 × 10^−9^ was used, the age of the WGD event with a peak value of approximately 1.2 was dated at 93–95 mya (*Ks* = 1.13–1.16) in the diploid and 96–101 mya (*Ks* = 1.17–1.23) in the triploid, with a confidence interval of 95%.

The *Ks* distribution peaks method can introduce inaccuracies because it relies on the assumption of a strict molecular clock and synonymous substitution rates, and synonymous substitution rates are usually variable across different species. To improve the accuracy of our results, we further performed absolute dating of the diploid via phylogenomic analysis to infer the age of the WGDs. Using this method, we obtained an absolute age distribution with a clear peak at 52 ± 1 mya with a confidence interval of 95% (Figure 3). We can also obtain the silent substitution rate r of 11.04 × 10^−9^ if combing *Ks* and absolute ages of *C*. *thalictroides*. As mentioned above, the evolutionary rate of *Ceratopteris* should be consistent, so we think the modified *Ks*/year rate is more accurate than a plant average *Ks*/year rate of 6.1 × 10^−9^, which could be applicable in the calculation of the date for other *Ceratopteris* species. Thus, we dated the same WGD in the triploid at 54 mya.

### 2.6. Phylogenomic Analysis of MADS-box Family Genes

MADS-box family genes, especially MIKC-type proteins, are well known to be involved in the developmental and morphological novelties in the sporophytic generation [31]. We identified and characterized a total of 32 candidate MADS-box genes in *C. thalictroides* (Appendix A), 27 of which were type II MADS-box genes. Our phylogeny of type II MADS-box genes revealed that three genes were clustered together with MIKC*-type genes (Figure 4). CRM1-, CRM3-, and CRM6- formed separate subclades that clustered with MIKC^c^ proteins in *C. thalictroides*. The CRM6 subclade consisted of a high number of type II MADS-box genes in *C. thalictroides*; however, only two new genes were identified in the CRM3 subclade, and these grouped together with AtAGL15 protein. A total of four genes, c24583_g4_i1, c16672_g3_i1, c24583_g1_i2, and c29048_g3_i1, were retained following the WGD event in *C. thalictroides*. These four genes grouped together with the previously reported CMADS3, CRM4, CMADS2, and CMADS4 proteins, respectively.

## 3. Discussion

### 3.1. Inferring WGD from Ks Age Distribution

WGDs are common in plants and are considered major factors driving plant diversity [3,5]. Several methods for the detection of WGDs have been reported, including synteny and gene-family trees with a reference species tree, and are widely used in plants with complete genome information. Another commonly applied method uses transcriptome data with a partial expressed sequence tag (EST), which is typically used in determining WGD features on the basis of *Ks* age distribution [3,6,9,12]. However, the quality of transcriptome data assembly significantly influences the identification and understanding of WGDs. In this study, we performed *Ks* analyses to detect the WGD events that have occurred in *C*. *thalictroides* and discussed the effect of polyploidy level on WGDs.

Nakazato et al. (2006) via genetic linkage mapping indicate that no ancient WGD events were detected in *C. richardii* [32], or signals of WGDs have been masked by chromosomal rearrangements and smaller-scale duplications. Nevertheless, multiple-copy genes and significant clustering observed in the linkage maps suggest that *C. richardii* may experience WGDs. Previous research has described a WGD event for *C. richardii*; however, the conclusion was built on low-coverage EST sequences with 631 duplicate pairs [24,33]. In our study on *C*. *thalictroides*, we used 8364 and 3088 paralogous pairs from diploid and triploid samples, respectively (Table 3), thus high-quality transcriptome assembly was possible. Our data is more reliable than previous studies because it covers a total of 69.6% (containing 63.7% complete orthologs and 5.9% partial sequence information) and 46.5% (with 34.3% complete orthologs and 12.2% fragmented genes) of the BUSCO sequences in the diploid and triploid, respectively (Table 2).

The *Ks*-based age distribution had an obvious peak near 1.2 in the diploid and the triploid samples of *C*. *thalictroides* (Figure 2), indicating that an ancient WGD event occurred in *C*. *thalictroides*. We also observed some additional small peaks after 1.2 for each sample. These results indicate that *C*. *thalictroides* might have undergone one or more WGD events. However, considering the stochastic effect and *Ks* saturation, which might lead to increasing uncertainty in *Ks* and artificial peaks in the distribution, we disregarded peaks with values larger than 2 [34], although there were seven and five peaks in the diploid and triploid, respectively (Table 3). The third additional component before 1.2 in the diploid might be the results of a small-scale duplication or stochastic variation because of its lower peaks. Peaks corresponding to young polyploidizations are often superimposed upon older large-scale duplication events such as WGDs [24]. The difference of the *Ks* frequency distribution and the peak value of 1.526 (diploid) and 2.268 (triploid) in Table 3 might be related to the effect of individual variability with the small sample size or the homoeologous recombination following hybridization and polyploidization during the formation of the triploid. The formation of triploids is still unknown, thus additional studies should be conducted in the future. In addition, it is unclear whether the WGD observed at the peak located at 1.526 and 2.268 are true WGDs or stochastic variations, thus additional research should be conducted. Additionally, although the *Ks* frequency distribution of the diploid significantly differed from that of the triploid, the ancient WGD event detected is strongly supported, suggesting that ancient occurrences of polyploidy had a larger impact on the *Ks* age distribution than on recently formed polyploids, which can be disregarded.

Our results reveal that *C*. *thalictroides* has indeed experienced an ancient WGD, as indicated by the obtained *Ks*-based age distributions. This event, which is prevalent but not limited to angiosperms, occurred more than once, and the recent polyploidy event has influenced *Ks* frequency distribution, but its weak interference on the detection of WGDs can be ignored.

### 3.2. Absolute Dating of the WGD Event in C. thalictroides

There are several different methods that can be used to estimate the age of a WGD event. In this study, we used both *Ks* distribution peaks and the absolute dating approach to infer the dates of WGD events of *C*. *thalictroides*. Our *Ks* distribution peak analyses dated the ancient WGD event at 93–101 mya based on the assumption that *C*. *thalictroides* experienced a similar evolutionary rate with angiosperms. However, our results may not be completely accurate because absolute ages converted by *Ks* value are more dependent on the assumption of a strict molecular clock and the synonymous substitution rates used. Synonymous substitution rates are variable among different plants, and the rates are unknown for fern species [10,35]. Although the *Ks* values and absolute ages of *E*. *giganteum* can be converted to synonymous substitution rates, the date estimation in *E*. *giganteum* is largely uncertain [9].

Therefore, to either confirm or refute the dating we obtained for the WGD event of the diploid based upon our *Ks* analysis, we further dated the WGD event using an absolute age distribution, which is a combination of the *Ks*-based relative age distribution and phylogenetic analysis. This method uses a relaxed clock model that assumes a lognormal distribution of evolutionary rates, is based on several fossil calibrations from a broad taxonomic sampling, and the age estimates for all actual duplicates present an approximate lognormal distribution, which is more likely to obtain accurate estimates than *Ks* distribution peaks. Absolute age distribution analyses have been widely used in many studies [9,10,36]. Figure 3 shows that the WGD event occurred at 52 ± 1 mya (95% confidence interval), which is dated at 54 mya in the triploid based on the assumption that *C*. *thalictroides* shared similar evolutionary rates; the small confidence interval represents a small chance of error. Therefore, we trust the results from the absolute age distribution analysis and placed the WGD event at approximately 52 mya, which is much closer to the PETM than the KT boundary [30]. In previous studies, WGDs were also identified in other ferns, including *E*. *giganteum*, *C. richardii* (180 mya), *Azolla*, and an earlier WGD shared by core leptosporangiates [9,23,24]. A WGD event in *E*. *giganteum* was estimated to be 75.16–112.53 mya (with a large confidence interval via absolute dating), and the WGD event placed at 50–70 mya using a *Ks* distribution based on the assumption that *E*. *giganteum* shares similar evolutionary rates with angiosperms [9]. In the case of *Azolla* and core leptosporangiates, no information was provided for the age of the WGD [23]. We obtained the silent substitution rate r to be 11.04 × 10^−9^ for the combination of *Ks* and absolute ages of *C*. *thalictroides*, which is higher than the average rate for plants (6.1 × 10^−^^9^) and cereals (6.5 × 10^−^^9^) and slightly lower than the range for *Drosophila* (15.6 × 10^−^^9^) and dicots (15 × 10^−^^9^) [6,10,25]. The rapid substitution rate might be linked with the life history of *C*. *thalictroides*, which is an annual and has a short generation time [27,37].

### 3.3. WGDs Contribute to Evolutionary Success and Potential Adaptation of Ceratopteris

Polyploidy is usually considered to be an evolutionary dead end. However, WGDs might occur at specific times, such as during dramatic climate changes, providing novel opportunities for evolutionary success [5]. The occurrence of WGDs are often correlated with mass extinction events and global climate changes [5,10,12]. Many WGD events, including several seed plant species, *E*. *giganteum*, and *p*. *patens*, are indicated to cluster around the KT boundary (60–70 mya), which led to the extinction of approximately 60% of the plant species on Earth [9,10,36]. However, the WGD event that occurred in *C*. *thalictroides* was inferred at 51–53 mya, which is closer to the PETM than the KT boundary. As is common in other tropical species, *C*. *thalictroides* may be more susceptible to increases in temperature, which facilitates the occurrence of WGD during the PETM [11,13,30,38,39]. Cenozoic climate issues occurring during the PETM, Early Eocene Climatic Optimum, and Mid-Eocene Climatic Optimum have intrigued scientists for a long time because of the drastically fluctuating temperature and atmospheric CO_2_ concentration. Viewed as a representative of extreme cases, PETM, also known as the Late Paleocene Thermal Maximum (55 mya), reached exceptionally warmer and wetter climates than any other period on Earth [30,40]. During the PETM, the global temperature increased by at least 5–10 °C due to massive carbon release.

Several studies have suggested that polyploids often have a greater chance to survive an extreme environment compared to diploids because they can undergo rapid morphological innovation [5,22,41,42,43,44]. As a pantropical genus, *Ceratopteris* displays greater heat tolerance and intolerance to cooler temperatures than the temperate taxon [45]. In *C. thalictroides*, we discovered that genes were retained after WGDs and these were not random but preferentially retained. Genes preferentially retained after the WGD related to its tolerance to contribute to its adaption to extreme environments [5,36]. Appendix A shows that genes related to response to stimuli, such as response to salt stress (GO:0071472) and heat acclimation (GO:0010286), were enriched, which is consistent with the climate changes during the PETM (55 mya), as well as higher temperature tolerance and the aquatic environments for extant *C. thalictroides* [30,38,40]. Previous reports have indicated that duplicates in the copy number alternation after WGDs have become subfunctionalized or neofunctionalized, contributing to the species diversity and evolutionary success [5,25]. We observed that four MADS-box proteins belonging to the type II CRM1 subclade that were retained following WGD in *C. thalictroides* (Figure 4), and a member of this gene family, CMADS4 (the homolog of c29048_g3_i1), might have acquired a novel function via neofunctionalization in root development to better survive in aquatic semi-aquatic environments. In addition, changes in copy number would lead to changes in gene expression, likely resulting in hybrid vigor and giving rise to genes and alleles available for selection, further leading to the development of an adaptive phenotype [5,46,47]. WGDs also contribute to the formation of the MIKC-type proteins in seed plants, most of which have existed in the common ancestor of fern and extant seed plants with ubiquitous expression, and were recruited to specific tissue giving rise to novel function such as floral development [48,49,50]. Changes in gene expression have also been shown to occur in allopolyploid cotton, which indicated a higher potential adaption than its diploid progenitors [5,51].

## 4. Materials and Methods

### 4.1. Chromosome Counting

Two samples of *C. thalictroides* were used in this study. One was collected from Guangdong Province, China, and cultivated in a greenhouse. This sample was previously used for transcriptome analysis [29]. The other sample was obtained from a greenhouse in Sun Yat-sen University, Guangdong, South China, the transcriptome of which was sequenced by Zhang et al. (2016) [28]. The young root tips of the sporophytes were pretreated in 0.002 mol/L 8-hydroxyquinoline solution for 3–6 h and then fixed in Carnoy’s solution (95% ethanol: glacial acetic acid = 3:1) for 12–24 h. Then, the samples were hydrolyzed at 37 °C with a mixture of 2% cellulase and pectinase for 1 h. They were then stained with carbol fuchsin. The chromosomes in the samples were counted and photographed using a Carl Zeiss Axio Scope A1 photomicroscope (Jena, Germany).

### 4.2. Data Collection

In total, transcriptome data from nine species was obtained. For the diploid sample of *C. thalictroides* and seven other fern species including *Goniophlebium niponicum*, *Woodwardia prolifera*, *Dennstaedtia pilosella*, *Cheilanthes chusana*, *Acrostichum aureum*, *Osmolindsaea odorata*, and *Alsophila podophylla*, where the candidate coding sequences and protein sequences were downloaded from *GigaScience* repository, *Giga*DB [29]. In the case of the triploid sample of *C. thalictroides*, the unigenes were obtained from the published data with accession number GEEK00000000 [28], and the candidate open reading frames of each putative unigene were predicted with TransDecoder (http://transdecoder.sf.net) (access on 1 August 2018) [52].

### 4.3. BUSCO Analysis

BUSCO analysis was employed for the evaluation of the completeness level of the transcriptome assembly [53]. Both sets of the *C. thalictroides* unigenes were blasted against a core set of conservative orthologs in plant species from the OrthoDB database (www.orthodb.org) (access on 10 August 2018), and the number of complete and partially matched genes were recorded.

### 4.4. Evaluation of WGD candidates on the Basis of Ks Age Distribution

To examine candidate WGD events in *C. thalictroides*, the *Ks*-based age distribution was performed as described previously [54]. An all-against-all BLASTP search for each transcriptome was performed using BLASTP version 2.2.29 (NCBI, Bethesda MD, USA) for the identification of gene families with a cutoff E-value of 1 × 10^−5^. A best match was considered significant if the alignment length was >100 amino acids and the expected value (E) was <1 × 10^−15^. Then, all pairs of sequences within each gene family were aligned using MUSCLE (v3.8.31, EMBL-EBI, Hinxton, Cambridgeshire, UK) with default parameters [55]. *Ks* estimates were generated through maximum likelihood estimation with the CODEML program of PAML version 4.8 (University College London, London, UK) [56]. A gene family of n members originates from n − 1 retained single gene duplications, whereas the number of possible pairwise comparisons within a gene family is n(n − 1)/2. To correct for the redundant *Ks* values, all *Ks* estimates for a particular duplication event were added to the *Ks* distribution, while the total weight of a single duplication event sums up to one [6]. The applied Python script is available online (http://github.com/EndymionCooper/KSPlotting). *Ks* values from 0.1 to 5 were retained for subsequent analyses, and Gaussian mixture models were performed in the R package mclust (v5.3, University of Washington, Seattle, WA, USA) for the fitting of log-transformed *Ks* distributions [9,57]. The age of a duplication event was inferred by using the relative divergence of the duplicates and the formula divergence date = *Ks*/(2 × r) if the number of silent substitutions increased approximately linearly with time. The plant average *Ks*/year rate of 6.1 × 10^−9^ was applied to the WGDs of *C*. *thalictroides* [25]. Additionally, the *Ks* frequency distributions of the diploids were compared with the triploid through the Wilcoxon matched pairs test [58].

### 4.5. Absolute Dating

Absolute dating for the WGD was performed to determine the age of the event, similar to the analysis described by Vanneste et al. (2014) [36]. A diagram of the work flow is provided in Appendix A. We determined the age of the WGD event using the diploid sample with a higher quality for this study. Duplicate pairs from the WGD peak with *Ks* values between 0.9 and 1.5 in the *Ks* distribution were collected. This resulted in 737 gene families. For each gene family, we retained the duplicates pair nearest the WGD peak boundaries as the representative homeologous pair, as multiple paralogous pairs might descend from the same gene duplication [36]. The orthologs were determined using a BLASTP analysis with the reciprocal best hit based on the protein dataset of peak-based duplicate pairs of the diploid *C*. *thalictroides* and seven other ferns, with an E-value of 1 × 10^−10^ [59]. Based on one homeologous pair, an orthogroup was obtained that contained the homeologous pair and several orthologs from other fern species (Appendix A), including *G. niponicum*, *W. prolifera*, *D. pilosella*, *C. chusana*, *A.*_*aureum*, *O. odorata*, and *A. podophylla*. A total of 706 orthogroups were collected. Maximum-likelihood trees were constructed using RAXML (Heidelberg Institute for Theoretical Studies, Heidelberg, Germany) under the GTRGAMMA model with 100 bootstrap replicates, and the paralogous pair of *C*. *thalictroides* that grouped together was used for further analyses. After removal of the orthogroups with gene trees in conflict with the species tree according to the Pteridophyte Phylogeny Group I system [60], a total of 153 orthogroups were obtained for the dating using BEAST v1.8.4 (University of Auckland, Auckland, New Zealand; University of Edinburgh, Edinburgh, UK; University of California, California, USA) with an uncorrelated relaxed clock model and a JTT+G; (four rate categories; Jones-Taylor-Thornton model and a site heterogeneity model gamma) evolutionary model [61]. Fossil calibrations were employed as follows: *C. chusana*, as the Pteroids, was dated to 93.5 mya, and *W. prolifera*, as *Woodwardia* (family *Blechnaceae*), was dated to 55.8 mya [62,63,64]. The chain length of the Markov chain Monte Carlo for each orthogroup was set to five million, with sampling every 1000 generations, thus a total of 200 samples. Tracer v1.5 (build-in BEAST 1.8.4) was used to estimate the trace file for all orthogroups, and the orthogroup was accepted if the minimum effective sample size was at least 100 [61]. A total of 153 orthogroups were accepted and all age estimates for the node adding homeologous pair were grouped into one absolute age distribution. The mclust package in R was applied to fit a mixture model based on the given grouped WGD age estimates, and a 95% confidence interval for the significant component was also estimated [57].

### 4.6. Functional Enrichment

As reported previously, genes are preferentially retained after WGDs [65,66]. We performed a functional enrichment analysis to determine whether biased functional retention in *C*. *thalictroides* occurs after a WGD and to understand whether the WGDs might be linked with extreme environmental changes. We obtained duplicate pairs with *Ks* values between 0.9 and 1.5, which lie under the WGD signature peak in the age distributions. GO enrichment analyses were performed using agriGO v2.0 (China Agricultural University, Beijing, China) with the singular enrichment analysis tool at default settings for the Fisher test (*p* < 0.05 signifying statistical significance) [67].

### 4.7. Identification of the MADS-box Gene Family in C. thalictroides

To identify putative MADS-box family genes in *C*. *thalictroides*, BLASTP was performed to search the *C*. *thalictroides* protein transcript database with the *Arabidopsis* MADS proteins as queries. An E-value cutoff of 1 × 10^−5^ was used. Additionally, proteins with SRF-TF domains (PF00319) were obtained from the Pfam database, and the hidden Markov model (HMM) was also performed to identify new members of the MADS-box gene family. All candidate MADS-box genes were examined by using the Simple Modular Architecture Research Tool (http://smart.embl-heidelberg.de/) and conserved domain databases to verify and remove incomplete MADS-box domains. We also removed redundant sequences with an identity higher than 99%.

### 4.8. Phylogenetic Analysis

To estimate the evolutionary relationship of MADS-box family genes retained in *C*. *thalictroides* and others, all putative MADS-box family genes in *C*. *thalictroides* and *Arabidopsis* were aligned using ClustalX (v1.83, Toby Gibson EMBL, Heidelberg, Germany), and manually adjusted. A Bayesian phylogenetic tree was constructed using the program MrBayes (v3.2.6) with the following settings: mixed model, 10,000 generations, and a sampling frequency of ten. The run was stopped when the standard deviation of the split frequencies was below 0.01 through adding generations and sampling frequency continually. A total of ten million generations were run and tree sampling density was 10,000 generations [68]. The first 25% of samples were discarded as burnin. FigTree (v1.4.2, University of Edinburgh, Edinburgh, UK) was used to visualize and edit the consensus tree in Appendix A [69]. In addition, a Bayesian phylogenetic tree was also constructed based on type II MADS-box genes from *Arabidopsis*, *Oryza sativa*, *Physcomitrella patens*, *Selaginella*
*moellendorffii,* and *C*. *thalictroides* (Figure 4), including the type II MADS-box genes previously isolated from *Ceratopteris*. *CgMADS1* from the streptophyte green alga, *Chara globularis*, was used as the outgroup.

## Figures and Tables

**Figure 1 ijms-20-01926-f001:**
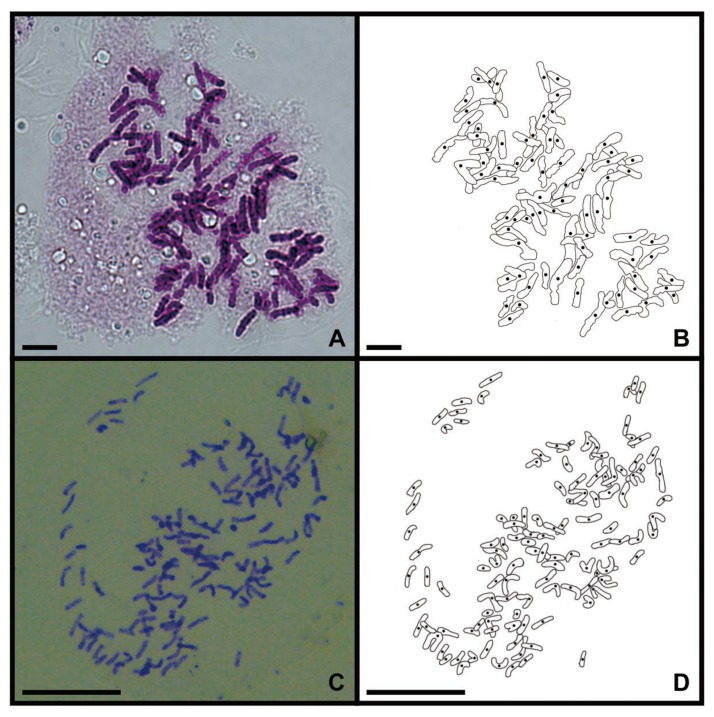
Chromosomes of *C. thalictroides* in mitotic root-tip cells (scale bars=20 μm). (**A**) Metaphase chromosome of the diploid, 2*n* = 78; (**B**) Lined drawing of Figure A; (**C**) Metaphase chromosome of the triploid, 2*n* = 117; (**D**) Lined drawing of Figure C.

**Figure 2 ijms-20-01926-f002:**
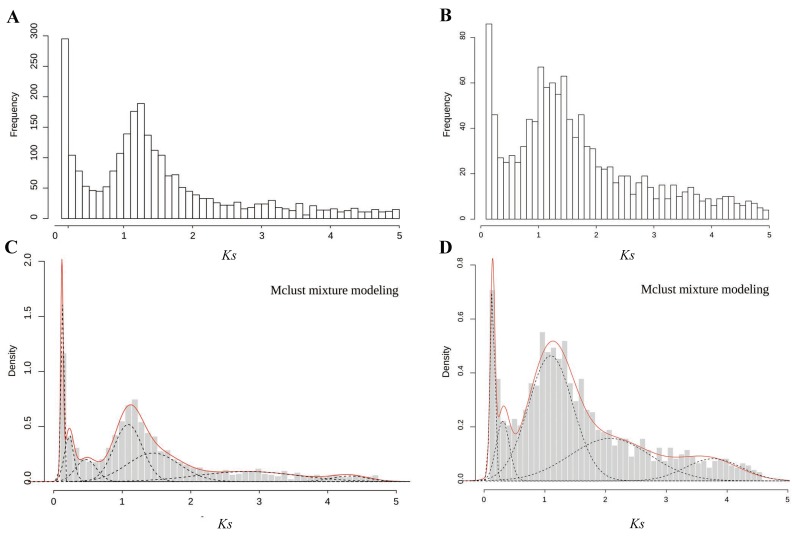
Frequency distributions of *Ks* values based on paralogous pairs of *C. thalictroides*. (**A**,**C**) Distributions of *Ks* values pairs of the diploid and triploid within 5. The x-axis represents the synonymous substitutions with a *Ks* cutoff of five in bins of 0.1, and the y-axis shows the number of retained duplicated paralogous gene pairs. (**B**,**D**) Mclust Gaussian mixture model analysis of (**A**,**C**), respectively. Optimal number of log-normal components overlaid on Ks distributions. The red line shows the sum of components.

**Figure 3 ijms-20-01926-f003:**
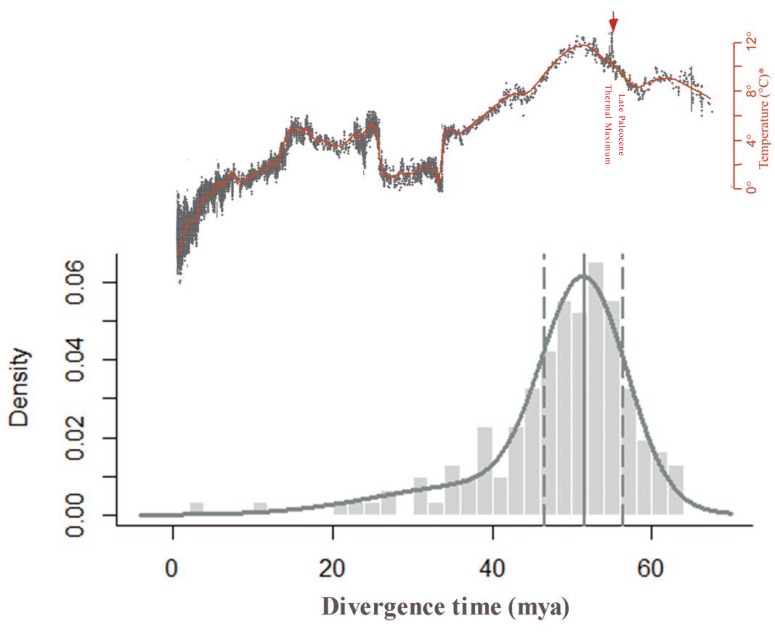
Absolute age distributions for the peak-based duplicates of *C. thalictroides* compared to the global climate changes during the Cenozoic. The vertical gray solid line represents its peak, viewed as the WGD age estimate, and the vertical gray dashed lines corresponded to 95% confidence intervals on the WGD age estimate. The parameters of statistically significant components identified using mclust were 52 mya ago and 0.76, which represent the inferred date and proportion, respectively. The global climate curve at the top comes from Zachos et al. (2001) with permission from AAAS [30].

**Figure 4 ijms-20-01926-f004:**
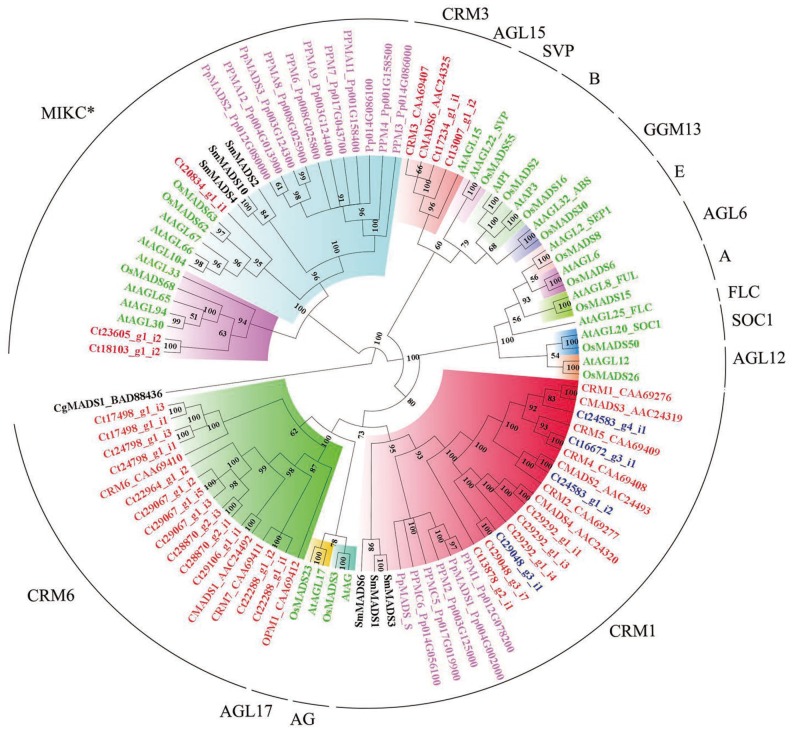
Phylogenetic tree of type II MADS-box proteins using MrBayes 3. The different clades are indicated by different colors, and *C. thalictroides* gene names are given in red, except for the four retained duplicates, denoted with blue, following WGD. Posterior probabilities are also indicated on the branches. The plant species included are as follows: *Arabidopsis*, *Oryza sativa* (Os), *Selaginella moellendorffii* (Sm)*, Physcomitrella patens* (Pp), *Chara globularis* (Cg), and *C. thalictroides* (Ct).

**Table 1 ijms-20-01926-t001:** A summary of the sequencing and assembly for diploid and triploid samples of *C. thalictroides* and seven other fern species.

Species	Total Reads (Clean)	Number of Contigs	Total Number of Unigenes	N50 (bp)	Mean Length (bp)
Triploid ^a^	35,528,634	69,929	60,823	787	576.00
Diploid ^b^	31,741,082	74,728	83,202	1610	912.26
*Goniophlebium niponicum* ^b^	38,786,214	54,152	58,494	1663	951.92
*Woodwardia prolifera* ^b^	40,967,322	69,931	74,564	1557	859.72
*Dennstaedtia pilosella* ^b^	45,618,446	84,813	89,185	1582	831.56
*Cheilanthes chusana* ^b^	51,851,066	49,449	52,782	1727	1012.63
*Acrostichum aureum* ^b^	43,422,574	46,189	50,594	1729	1043.2
*Osmolindsaea odorata* ^b^	46,808,646	113,778	130,549	1521	845.96
*Alsophila podophylla* ^b^	48,768,608	66,254	72,404	1580	904.62

Note: ^a^ refer to the triploid *C. thalictroides* [28], ^b^ refer to the diploid *C. thalictroides* and the other seven fern species [29].

**Table 2 ijms-20-01926-t002:** BUSCO results (genome completeness) for diploid and triploid samples of *C. thalictroides*.

Species	BUSCO Notation Assessment Results
Diploid	C: 63.7% [S:39.7%, D:24%], F:5.9%, M: 30.4%, n: 1440
Triploid	C: 34.3% [S:30.5%, D:3.8%], F:12.2%, M: 53.5%, n: 1440

BUSCO was used to assess the transcriptome data quality with 1440 conservative orthologs in plant species as reference. Abbreviation: C—Complete Single-Copy BUSCOs; S—Complete and Single-Copy BUSCOs; D—Complete Duplicated BUSCOs; F—Fragmented BUSCOs; M—Missing BUSCOs; N—Total BUSCO groups searched.

**Table 3 ijms-20-01926-t003:** Mixture modeling of the age distribution of *C. thalictroides* presented in Figure 2.

No. of Duplicates	No. of Components	Bayesian Information Criterion	Mixture Means (*Ks*)	Variance (*Ks*)	Proportion
Diploid					
8364	7	5998.011	0.128	0.0004	0.091
8364	7	5998.011	0.238	0.0048	0.074
8364	7	5998.011	0.499	0.0248	0.079
8364	7	5998.011	1.148	0.0461	0.278
8364	7	5998.011	1.526	0.1619	0.265
8364	7	5998.011	2.989	0.6015	0.176
8364	7	5998.011	4.561	0.0790	0.036
Triploid					
3088	5	3380.834	0.154	0.0016	0.075
3088	5	3380.834	0.338	0.0135	0.066
3088	5	3380.834	1.199	0.1591	0.464
3088	5	3380.834	2.268	0.5133	0.282
3088	5	3380.834	4.019	0.2962	0.112

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
