# Peer review of "Dating Whole Genome Duplication in Ceratopteris thalictroides and Potential Adaptive Values of Retained Gene Duplicates"

_ijms, 2019, doi:10.3390/ijms20081926_

Reviewer 1 Report

ijms-474761

Dating whole genome duplication in Ceratopteris thalictroides and potential adaptive values of retained gene duplicates

This is an interesting study about whole-genome duplications (WGDs) occurred in a model fern species Ceratopteris thalictroides. Authors estimated the age of WGD event by using high-quality RNA seq data and they found that the estimated WGD age (about 52 mya) coincided with Paleocene-Eocene Thermal Maximum (PETM). Authors also performed a Gene Ontology enrichment analysis and phylogenetic analysis of certain genes, and showed that genes related stress-responses were enriched after WGD.

For the benefit of the readers, however, a number of points need clarifying. These are given below.

1. Dating the WGD event using Ks distribution peaks and absolute dating (section 2.5)

  Using a plant average Ks/year rate (6.1x10^-9), the age of the WGD was estimated at 93-95 mya. Authors then performed absolute dating of the diploid via phylogenetic analysis to improve the accuracy of the results (modified Ks/year rate = 11.04x10^-9). I think authors should explain the appropriateness of modified Ks/year rate in more detail (in 4. Materials and Methods, 2. Results).

2. References

  There are some papers concerning polyploidy and evolution of ferns not referred in this manuscript.

Nakazato T et al. (2006) Genetic map-based analysis of genome structure in the homosporous fern Ceratopteris richardii. Genetics 173 (3): 1585-1597.

Clark J et al. (2016) Genome evolution of ferns: evidence for relative stasis of genome size across the fern phylogeny. New Phytologist 210(3): 1072-1082.

Schneider H et al. (2017) Neo- and Paleopolyploidy contribute to the species diversity of Asplenium – the most species rich genus of ferns. Journal of Systematics and Evolution 55(4): 353-364.

I think information of these papers (introduction or discussion) help readers understand the results presented in this manuscript.

Author Response

Response to Reviewer 1 Comments

Point 1: Dating the WGD event using Ks distribution peaks and absolute dating (section 2.5)

Using a plant average Ks/year rate (6.1x10^-9), the age of the WGD was estimated at 93-95 mya. Authors then performed absolute dating of the diploid via phylogenetic analysis to improve the accuracy of the results (modified Ks/year rate = 11.04x10^-9). I think authors should explain the appropriateness of modified Ks/year rate in more detail (in 4. Materials and Methods, 2. Results).

Response 1: Thank you very much for your advices. As you suggested, we have added detail explanations of the appropriateness of modified Ks/year rate in the revised manuscript (Line 185-187, in 4. Materials and Methods, 2. Results)

Point 2: References

There are some papers concerning polyploidy and evolution of ferns not referred in this manuscript.

Nakazato T et al. (2006) Genetic map-based analysis of genome structure in the homosporous fern Ceratopteris richardii. Genetics 173 (3): 1585-1597.

Clark J et al. (2016) Genome evolution of ferns: evidence for relative stasis of genome size across the fern phylogeny. New Phytologist 210(3): 1072-1082.

Schneider H et al. (2017) Neo- and Paleopolyploidy contribute to the species diversity of Asplenium – the most species rich genus of ferns. Journal of Systematics and Evolution 55(4): 353-364.

I think information of these papers (introduction or discussion) help readers understand the results presented in this manuscript.

Response 2: Thank you so much for your advices. The first two references have been added in introduction in Line 67, and the later two were added in Reference section (Line312, Line226).

Reviewer 2 Report

Specific comments referring to line numbers, tables or figures. Reviewers need not comment on formatting issues that do not obscure the meaning of the paper, as these will be addressed by editors.

1.Introduction is nicely and understandable written.

2. Results:

2.1. Chromosome counting

Line 105: The authors should put the Figure 1. after this line (the results of chromosome counting)

Line 111: The authors should let the Table1. here, but it should be better to start a new paragraph with the 2.2. BUSCO analysis (with Table 2. below).

Line 117: In Table 1. the authors should indicate in the Reference that the Ref27 belongs to the diploid C. thalictroides. This reference is missing here.

Line 118: More explanation is necessary for Table 2 in the Legends (not only abbreviations).

Line 153: in the text Table S1 can be found but in the Supplementary data this is written like Table S2 (on the top of the columes)

Line 153: in the text „a total of 11 GO terms were found”, but the the Table S2 contains 113 such GO terms.

Table S2:

Legend: „All listed terms are significantly enriched for the genes retained after WGD by Fisher's exact test (p=0.05) with a false discovery rate (FDR) correction. BP, Biological process; CP, cellular component; MF, molecular function.” but in the Table 2 inside there is CC = cellular component (I guess this is the right one)

Line 154: write in the text that biological processes (BP)

Line 162: write in the text that cellular component (CC)

Line 165: For me the Figure 3 looks like very small, and I would prefer to put it into the Supplementary data.

Line 165: Figure 3. legend: please, write the species name at the end of the sentence.

Line 190: In Figure 4. legend the Zachos et al (2001) has a wrong Reference number, it is Ref56, not Ref52.

Line 194: Figure S3 = Image3 in the Supplementary Material, you should rename it as Figure S3. Anyhow, this is the first Supplementary Figure what I saw untill now in the text, therefore it should be called as Figure S1.

Line 204: In Figure 5. legend the abbreviations are missing like Arabidopsis thaliana (At) and C. thalictroides (Ct).

3.Discussion: OK

4. Material Methods

Line 334: Carl Zeiss Microscope … please, write where it is arising from (city, country)

Line 372: „A diagram of work flow is provided in Figure S1.” This is again indicated in the Supplementary Material as an Image 1. Logically, this should be the Figure S2. Please, correct to Figure S2 in the Supplementary Material part.

Line 381: „several orthologs from ferm species (Figure S2)”.  This is again indicated in the Supplementary Material as an Image 2. Logically, this should be the Figure S3. Please, correct to Figure S3 in the Supplementary Material part.

Line 382. Instead of D. pilosella (which is written in the text), there is D. hirsuta in the Supplemental Image 2. This latter species was not scored even in the Shen et al. (2018) paper, so I doubt that it was seqenced at all.

Line 424: „as burnin Fig 3 was used to visualize and edit the consensus tree”. This is indicated in the Supplementary Material as an Image 3. Logically, this should be the Figure S4. Please, correct to Figure S4 in the Supplementary Material part.

Line 425: ”In addition, a Bayesian phylogenetic tree was also constructed based on type IIMADS-box genes from Arabidopsis, Oryza sativa, Physcomitrella patens, Selaginella moellendorffii, and C. thalictroides,  including the type II MADS-box genes previously isolated from Ceratopteris. CgMADS1 from the streptophyte green alga, Chara globularis, was used as the outgroup.”  Where is the mentioned phylogenetic tree figure?

Author Response

Response to Reviewer 2 Comments

Point 1: Line 105: The authors should put the Figure 1. after this line (the results of chromosome counting)

Response 1: Thank you very much. As you suggested, it has been moved to Line 108 in the revised manuscript.

Point 2:  Line 111: The authors should let the Table1. here, but it should be better to start a new paragraph with the 2.2. BUSCO analysis (with Table 2. below).

Response 2: Thanks a lot. We have moved the table 1 in the revised manuscript (Line 117-118).

Point 3: Line 117: In Table 1. the authors should indicate in the Reference that the Ref27 belongs to the diploid C. thalictroides. This reference is missing here.

Response 3: Thank you so much. We have revised table1 to make it clearer (Line 117-120).

Point 4: Line 118: More explanation is necessary for Table 2 in the Legends (not only abbreviations).

Response 4: Thanks. We have added detail explanations for Table 2 as you suggested (Line 128).

Point 5: Line 153: in the text Table S1 can be found but in the Supplementary data this is written like Table S2 (on the top of the columes)

Response 5: Thanks for your comments. It is really not right in Table S1 in the supplementary data, and we have corrected it according to your comments.

Point 6: Line 153: in the text „a total of 11 GO terms were found”, but the the Table S2 contains 113 such GO terms.

Response 6: Thanks a lot. After checked, the Table S1 (Table S2 as you mentioned) contains 111 GO terms except the table head and the Legends, which is consistent with the data in the text.

Point 7: Table S2: Legend: „All listed terms are significantly enriched for the genes retained after WGD by Fisher's exact test (p=0.05) with a false discovery rate (FDR) correction. BP, Biological process; CP, cellular component; MF, molecular function.” but in the Table 2 inside there is CC = cellular component (I guess this is the right one)

Response 7: Thank you so much. It is really not exact in the Table S1: Legend, we have revised the sentences as you suggested.

Point 8: Line 154: write in the text that biological processes (BP)

Response 8: Thank you. This sentence has been changed in the revised manuscript (Line 161)

Point 9: Line 162: write in the text that cellular component (CC)

Response 9: Thanks a lot. This sentence has been revised according to your comments (Line 168).

Point 10: Line 165: For me the Figure 3 looks like very small, and I would prefer to put it into the Supplementary data.

Response 10: We are grateful to your suggestion. We have changed the Figure 3 to Figure S1 in the revised manuscript.

Point 11: Line 165: Figure 3. legend: please, write the species name at the end of the sentence.

Response 11: Thank you. Figure 3 has been changed to Figure S1, and we will add the species name in the figure legend according to your advice.

Point 12: Line 190: In Figure 4. legend the Zachos et al (2001) has a wrong Reference number, it is Ref56, not Ref52.

Response 12: Thanks. The reference related to the Zachos et al (2001) has been changed to Ref30 in the revised manuscript (Line 195).

Point 13: Line 194: Figure S3 = Image3 in the Supplementary Material, you should rename it as Figure S3. Anyhow, this is the first Supplementary Figure what I saw untill now in the text, therefore it should be called as Figure S1.

Response 13: Thanks for your comments. In the revised manuscript, Figure S3 has been changed to Figure S2 because the Figure 3 in the original manuscript has been changed to Figure S1.

Point 14: Line 204: In Figure 5. legend the abbreviations are missing like Arabidopsis thaliana (At) and C. thalictroides (Ct).

Response 14: Thank you for your advice. The abbreviations including Arabidopsis thaliana (At) and C. thalictroides (Ct) have been added in the Figure 4 and Figure Legend (Line 208-213).

Point 15: Line 334: Carl Zeiss Microscope … please, write where it is arising from (city, country)

Response 15: Thank you. This content has been added in the revised manuscript (Line 343).

Point 16: Line 372: A diagram of work flow is provided in Figure S1.” This is again indicated in the Supplementary Material as an Image 1. Logically, this should be the Figure S2. Please, correct to Figure S2 in the Supplementary Material part.

Response 16: Thanks a lot. Figure S1 in the original manuscript has been revised to Figure S3 according to your advice.

Point 17: Line 381: „several orthologs from ferm species (Figure S2)”.  This is again indicated in the Supplementary Material as an Image 2. Logically, this should be the Figure S3. Please, correct to Figure S3 in the Supplementary Material part.

Response 17: Thank you. We have revised Figure S2 to Figure S4 in the revised manuscript as you suggested.

Point 18: Line 382. Instead of D. pilosella (which is written in the text), there is D. hirsuta in the Supplemental Image 2. This latter species was not scored even in the Shen et al. (2018) paper, so I doubt that it was seqenced at all.

Response 18: Thank you so much. D. pilosella is the synonym of D. hirsuta, which has been revised to D. hirsuta in Flora of China (http://foc.iplant.cn/content.aspx?TaxonId=242317408). To be consistent with the paper Shen et al. (2018), we have replaced D. hirsuta to D. pilosella in the Figure S4 in the revised manuscript.

Point 19: Line 424: „as burnin Fig 3 was used to visualize and edit the consensus tree”. This is indicated in the Supplementary Material as an Image 3. Logically, this should be the Figure S4. Please, correct to Figure S4 in the Supplementary Material part.

Response 19: Thank you. Figure S3 has been changed to Figure S2 in the revised manuscript.

Point 20: Line 425: ”In addition, a Bayesian phylogenetic tree was also constructed based on type IIMADS-box genes from Arabidopsis,Oryza sativa, Physcomitrella patens, Selaginella moellendorffii, and C.thalictroides, including the type II MADS-box genes previously isolated from Ceratopteris. CgMADS1 from the streptophyte green alga, Chara globularis, was used as the outgroup.” Where is the mentioned phylogenetic tree figure?

Response 20: Thanks a lot. We have added Figure 4 according to your advice in the revised manuscript (Line 434).